# Differences between Sexes and Speed Levels in Pelvic 3D Kinematic Patterns during Running Using an Inertial Measurement Unit (IMU)

**DOI:** 10.3390/ijerph20043631

**Published:** 2023-02-18

**Authors:** Sara Perpiñá-Martínez, María Dolores Arguisuelas-Martínez, Borja Pérez-Domínguez, Ivan Nacher-Moltó, Javier Martínez-Gramage

**Affiliations:** 1Department of Nursing and Physiotherapy Salus Infirmorum, Universidad Pontificia de Salamanca, 37002 Madrid, Spain; 2Department of Nursing and Physiotherapy, Universidad Cardenal Herrera CEU, CEU Universities, 46115 Valencia, Spain; 3Department of Physiotherapy, Universidad de Valencia, 46010 Valencia, Spain; 4Head of Human Motion & Biomechanics in DAWAKO Medtech, Faculty of Medicine and Health Sciences, Catholic University of Valencia, 46001 Valencia, Spain

**Keywords:** biomechanics, kinematics, pelvis, running, wearables, exercise

## Abstract

This study aimed to assess the 3D kinematic pattern of the pelvis during running and establish differences between sexes using the IMU sensor for spatiotemporal outcomes, vertical acceleration symmetry index, and ranges of motion of the pelvis in the sagittal, coronal, and transverse planes of movement. The kinematic range in males was 5.92°–6.50°, according to tilt. The range of obliquity was between 7.84° and 9.27° and between 9.69° and 13.60°, according to pelvic rotation. In females, the results were 6.26°–7.36°, 7.81°–9.64°, and 13.2°–16.13°, respectively. Stride length increased proportionally to speed in males and females. The reliability of the inertial sensor according to tilt and gait symmetry showed good results, and the reliability levels were excellent for cadence parameters, stride length, stride time, obliquity, and pelvic rotation. The amplitude of pelvic tilt did not change at different speed levels between sexes. The range of pelvic obliquity increased in females at a medium speed level, and the pelvic rotation range increased during running, according to speed and sex. The inertial sensor has been proven to be a reliable tool for kinematic analysis during running.

## 1. Introduction

Running is one of the most popular and accessible activities for the population [1], and its popularity has grown exponentially in the last 50 years [2]. Consequently, there has been an increase in rates of injury, especially in beginners lacking experience, with up to 30% of new runners affected every year [3]. Moreover, the repetitive nature of running makes it an activity with a high injury risk [4], which ranges from 3.2% to 84.9% [5,6], with a median prevalence of 44.6% ± 18.4% [7]. Out of the many running injuries, 70–80% of them are caused by overuse [7], with Aquilian tendinopathy, plantar fasciitis and patellofemoral, iliotibial band, and the medial tibialis stress syndromes being the most prevalent [7].

Running injuries’ etiology is multifactorial, yet it is not possible to determine the exact cause for every injury because movement during running requires a precise inter-segmental coordination [8]. Among studied risk factors, previous injuries [9], high body mass index (BMI) [10], sex, age, experience [11], training alterations [3,9,12], biomechanical issues [13], and fatigue [14,15] are the most prevalent. It has been observed that injured runners change their movement pattern to prevent further damage [14]. Despite this, overall results turn out to be inconsistent due to the large number of potential outcomes that could direct runners to injury.

According to a biomechanical analysis carried out during running, the pelvis plays a stabilizing role and transfers energy between the lower extremity and the rest of the body [16], creating stress in the back [17] and distal structures of the lower limb [18] if there is an alteration in the coordination of pelvic and vertebral movements. The biomechanical analysis of running is an important way to assess movement in individuals with injuries, including simple spatiotemporal parameters and complex three-dimensional movements [19]. To achieve this, it is necessary to establish the role of the biomechanical coordination of the pelvis during running.

Three-dimensional (3D) optoelectronic systems are considered the Gold Standard in the analysis of movement, surpassing clinical observation [20,21]. Despite this, due to its relatively high cost and the time and space required to develop an analysis using these systems [22], in addition to the difficulty of analyzing certain planes in 3D, the technological progress has enabled the development of more affordable, accessible, and feasible devices [23]. An example of this is the inertial measurement unit (IMU), a portable, valid, and reliable device [24] that facilitates assessment of the orientation of the segments and the articular angles [25,26]. In addition, this device shows multiple correlation coefficients above 0.95 with respect to the Gold Standard when comparing different running speed levels [27], presenting high correlations for the angles of tilt, obliquity, and rotation [28,29].

Research studies, such as the one conducted by Novacheck [30], determine reference ranges for the pelvis, which are considered the normative standards for running patterns in the sagittal, coronal, and transverse planes of movement, through analysis with optoelectronic devices. However, in order to account for progress in technology, footwear [31], cultural variations [32], and an increase in physical activity levels in sedentary populations [33], among other factors, these ranges must be revised, particularly given the kinetic and kinematic biomechanical differences between males and females during running [34]. Analyses should also be established according to different speed levels to determine how the biomechanics behave in every type of runner.

Knowing the kinematic ranges of the pelvis at different speed levels during running and according to sex could help form an understanding of the potential role its alterations might have in functional or structural injuries, by determining how the pelvis interacts during each phase of the running cycle. The main objective of this study is to determine the 3D kinematic pattern of the pelvis during running and to establish differences between sexes using the IMU sensor for the spatiotemporal outcomes, gait symmetry index, and amplitude of motion ranges of the pelvis in the sagittal, coronal, and transverse planes of movement. A secondary objective is to determine the reliability of the IMU sensor for these variables.

## 2. Materials and Methods

### 2.1. Participant Charateristics

A total of 101 participants were included in the study to determine normative values: 51 males and 50 females. Ages ranged from 18 to 53 years, with a mean of 31.3 years. Mean weight was 65.7 Kgs and mean height was 170 cm (Table 1). Out of the initial 107 participants, 6 were excluded due to several reasons (medical criteria *n* = 1 and foot blisters *n* = 5). The analysis of the reliability of the IMU sensor included 29 participants, with a mean age of 31.2 years, a mean weight of 66 Kgs, and a mean height of 172 cm (Table 1).

Participants in this study were healthy subjects with no current injuries who had at least 1 year’s running experience, and who had trained for at least a total of 90 min, distributed across weekly training sessions. Participants were excluded if they were older than 65 years, had suffered an injury in the lower limb in the last year, had undergone a surgical intervention, or had neurological problems that might have altered the biomechanics of the standard running cycle pattern.

Recruitment took place through circulation in the electronic channels of the triathlon clubs of the Valencian Community and running teams. This study was approved by the Ethics Committee of the University CEU Cardenal Herrera, in Valencia (CEI 14/018), and was conducted according to the basic principles of the Declaration of Helsinki. Every participant was briefed regarding the nature of the study and was asked to give written consent to participate.

### 2.2. Procedure

In the first phase of the study, standardized values and differences between sexes were established involving the same outcomes.

The study then determined the reliability of the IMU sensor in the biomechanical analysis of running. Several spatiotemporal outcomes were assessed, including cadence, running cycle, stride length, and vertical acceleration symmetry index. Anterior–posterior tilt pelvic ranges, obliquity, and pelvic rotation amplitudes were also assessed in females and males.

The amplitude of the pelvis 3D movements and spatiotemporal outcomes were assessed using an inertial sensor BTS G-Sensor (BTS Bioengineering, Garbagnate Milanese, Italy) with an ergonomic belt at the height of S1 (Figure 1) to capture different kinematic and spatiotemporal outcomes. This IMU comprised a 16-axis triaxial accelerometer with multiple sensitivities (±2, ±4, ±6, ±8, and ±16 g) with a frequency of 4 Hz to 1000 Hz, a triaxial gyroscope with multiple sensitivities (±250, ±500, ±1000, ±2000 o/s), with a frequency oscillating between 4 Hz to 8000 Hz, and a triaxial 13-bit magnetometer (±1200 uT), with a frequency exceeding 100 Hz.

In this study, we used a treadmill (BH Fitness Columbia Pro 130 cm × 40 cm) to establish standardized conditions under which the kinematic outcomes of running would be more reproducible. We set the incline to 1° and allowed each participant to select the speed [34,35] at which they regularly trained (self-selected speed). The participants performed in their regular training shoes and were allowed a 5 min warm-up period to adjust to the treadmill. According to protocols used in previous running biomechanics studies [36,37], the initial speed was progressively increased over 2 min and was then maintained for 3 min while the data were collected. 

Spatiotemporal outcomes including cadence, stride length, and running cycle were registered. Vertical acceleration symmetry index, kinematic ranges of tilt, obliquity, and pelvic rotation, as well as the participant’s age, weight, height, running experience, and weekly training volumes were also registered. To determine the reliability of the sensor, participants performed 2 tests of 5 min each with a 30 min margin between them, under the same circumstances. Environmental conditions were 22–25° and relative humidity levels ranged from 40 to 55%. 

The participants who were assessed to establish normative values for pelvic kinematics during running using the inertial sensor were stratified into three groups: “Slow speed” was considered for values between 9.98 km/h and 8.75 km/h, “medium speed” was considered for values between 9.98 and 11.70 km/h and between 8.75 km/h and 10.11 km/h, and “fast speed” was considered for values above 11.71 km/h and 10.11 km/h for males and females, respectively.

### 2.3. Statistical Analysis

To describe the demographical data of the sample, descriptive statistics were independently calculated for each sex. Normality distribution was assessed for the independent outcomes using the Kolmogorov–Smirnov test and homogeneity of variance was assessed using Levene’s test. Regarding the reliability analysis, the Shapiro–Wilk test was used to determine the normal distribution of the sample.

Maximal and minimal values were identified at every speed level for both males and females and these were divided into 2 percentiles: 33 and 66. Participants were allocated to one of 3 groups between these percentiles: slow, medium, and fast speed.

To assess the reliability of the IMU sensor, the Intraclass Correlation Coefficient (ICC) was calculated.

Inferential statistics were performed according to sex and speed. A 2-way factorial ANOVA between subjects was used to determine the kinematic outcomes, the factors being sex and speed. The level of significance was established at *p* < 0.05 with a confidence interval of 95%. Statistical calculations were performed with the SPSS software in its 18.0 version.

Epidat 4.2 software was used to obtain the sample size of the 2 tests. In both cases, an alpha error of 0.05 was accepted with a power of 85%. To establish normative values, calculations were performed for two independent means, with a standard deviation of 1.68 [38] and a minimal detectable change of 1.1 units, requiring a minimum sample of 45 subjects in each group. To check the reliability of the sensor, calculations were performed for two dependent means, with a minimum sample of 25 subjects in total.

## 3. Results

### 3.1. Normative Values of the Pelvic Kinematics and Spatiotemporal Outcomes

Regarding kinematic ranges, the mean amplitude of pelvic tilt for males and females oscillated between 5.92° and 7.36° (Figure 2) without statistically significant differences, but with a tendency to increase as speed increased in females, whilst in males a bigger range was observed at medium speed (Table 2).

Mean pelvic obliquity oscillated between 7.84° and 9.64° (Figure 3). The obliquity in females at medium speed was significant with respect to males (*p* < 0.05). No statistical differences were found in the remaining speed level. Mean pelvic rotation ranged from 9.96° to 16.13° (Figure 4). Significant differences were found in speed (*p* < 0.05) and sex (*p* < 0.001) but not in the interaction between them. Pelvic rotation was higher in females with respect to males at every speed level. In addition, both sexes increased their pelvic rotation as their speed increased.

No statistically significant differences were found regarding spatiotemporal outcomes of cadence and running cycle according to sex, speed levels, or interaction. With regard to stride length, it was observed that it increased statistically as speed increased in both males and females, but with no differences in the interaction. The vertical acceleration symmetry index did not show differences according to sex and speed levels, but differences were found according to the interaction of sex and medium speed (*p* < 0.05), where males increased their symmetry with respect to females.

Finally, regarding the running cycle waveforms at different speed levels between sexes, anterior–posterior tilt range presented the same trace in both females and males at every speed level. However, the trace of the kinematics for pelvic obliquity at medium speed level was wider in females than in males. Lastly, the trace of the pelvic rotation in females and males did not present any differences. 

### 3.2. Reliability of the IMU Sensor during Running

The results obtained from the test–retest present high reliability scores for the IMU sensor in every analyzed outcome (ICC > 0.80) (Table 3). Reliability values for the symmetry index and pelvic tilt were >0.80, whilst for the remaining outcomes, they oscillated between 0.922 and 0.997, showing the highest correlation level in the spatiotemporal parameters of stride length, stride time, and cadence, respectively.

## 4. Discussion

This study identified a series of biomechanical outcomes that enabled the detection of the differences between sexes during running at different speed levels in regard to pelvic 3D kinematics and assessed the reliability of an IMU sensor for their analysis. To the best of our knowledge, this is the first study to show the differences in the kinematic pattern during running between men and women, as assessed through an IMU sensor. In particular, a kinematic reference standard was established which can be used in a clinical setting, enabling the use of a portable medical device to gain a biomechanical understanding of the pelvis during running.

A treadmill was used to control speed, slope, and terrain conditions as much as possible, following the protocol proposed by several authors [39]. Although there are studies that question using a treadmill because it increases hip flexion [40,41] and reduces stride length [42], recent reviews argue that kinematic assessments, kinetics, muscle activity, and spatiotemporal outcomes are comparable when running on a treadmill and on other surfaces [43]. Additionally, to avoid biases that might alter running biomechanics, such as fatigue [44], we opted for a self-selected speed level. Speed alterations are also closely related to biomechanical factors such as stability, lift time, and the contact time of the leg [37]. This is reinforced by the results found by Kong et al., in which it was concluded that self-selected speed levels eliminate abnormal kinematic patterns [45]. In addition, at different speed levels, the same participant changes their own running biomechanics [46].

Regarding the amplitude of tilt range, in our study it is stable at averages between 5.92° and 7.36° in self-selected speed levels. Although several studies assess the oscillation widths of the pelvis in the sagittal, coronal, and transverse planes of movement [30,47], only Schache found differences in 7.8° and 9.4° for males and females, respectively. The study conducted by Novacheck shows an average amplitude tilt of 5° [30]. A force absorption mechanism occurs in the pelvis with initial contact, creating a slight posterior tilt with a minimal lumbar flexion and a slight hip flexion [48]. Following that, the pelvis initiates a posterior tilt movement to absorb the load, followed by an anterior tilt that will have its maximum peak just after the propulsion phase. However, although the maximum peak occurs in this phase, the inversion of the angular movement of the pelvis may be delayed due to a reaction after propulsive speed at high-speed levels, occurring after the take-off phase at low-speed levels [30]. An alteration in tilt is related to muscle injuries in the hamstrings, due to their biarticular nature; more so in the swing phase when stride length can be altered [49]. An increase in kinematic tilt range might reduce the power of propulsion in the athlete.

During running, pelvic obliquity is produced when the muscle contracts eccentrically to absorb landing forces, as well as to provide stability and efficiency [30]. The increase in pelvic obliquity, known as pelvic drop, has been widely related to musculoskeletal pathologies such as iliotibial band syndrome [50], shin pain [51], and medial patellofemoral syndrome [52,53]. Due to a major hip adduction and a greater peak knee valgus [30], it could possibly be related to pelvic morphology or a deficit in abductor muscle activation in women, forcing the knee and the hip to increase energy absorption [34], increasing the tension in the internal compartment of the knee and the iliotibial band during landing phase [53], or increasing the pronation of the foot [51].

The obliquity ranges observed in our study differ from those reported by other authors. A considerable difference is observed when comparing our results to the ones presented by Schache, which established obliquity at 13.8° and 19.3° for men and women, respectively [47]. In our study, those ranges are stable, with the exception of medium speed levels between sexes, which were higher in women. The muscle responsible for counteracting hip adduction and pelvic inferior obliquity is the medial gluteus, which is considered to be the muscle in charge of pelvic horizontality [54]. In females, when abductor muscles should be working eccentrically in the activation, offering a counteraction in the descent of obliquity, there is an activation delay, producing an increase in the articular range of movement. Increases in mobility in the sagittal and frontal planes may suggest a lack of stability in the lumbopelvic complex and deficient energy transmission towards the lower limbs in the propulsion phase, accompanied by an excess of cushioning when responding to loading.

Regarding pelvic rotation outcomes, significant differences were observed between sexes, with an average range of rotation from 13.21° to 16.13° for females and from 9.96° to 13.60° for males. Comparing these results to the reference studies on the assessment of pelvic dynamics, our results are inferior to those found in males, which range from 16° to 18°, and it can be hypothesized that these changes are due to speed [30,55]. During running, when toes are lifted in the propulsion phase, the pelvis presents its maximum tilt, slight homolateral obliquity in the support, and slight external rotation of the hip, also producing hip flexion limitation, which could end up being the reason for increased stride length. This increase in pelvic rotation in females is related to a genetic predisposition to be more flexible [56,57] and to have reduced elastic energy storage [58], which is associated with a reduction in the peak maximal force [59], requiring this compensation at every speed level. This increase in rotation, in addition to pelvic morphology in women, constitutes the main biomechanical difference between sexes in controlling alignment between the hip and the knee. According to Ferber et al. [34], a greater probability of suffering lower limb injuries is established with respect to men during running, presenting a relative risk ratio of 2.4 [60]. Women present different biomechanical functioning during the landing phase, which is characterized by a higher maximum peak adduction and internal rotation of the hip and external rotation of the knee [34]. In this phase, as shown in Figure 4, at around 5–10% of the running cycle, we can observe maximal peak external rotation of the pelvis in women, coinciding with maximal peak vertical ground reaction force, a moment that requires higher stability levels at the lower limb.

Due to the elevated clinical impact of running, the secondary objective was to demonstrate the reliability of the IMU on the pelvic 3D kinematic during running; ICC values above 0.8 were obtained in every observed outcome, which shows high to excellent reliability levels.

Generally, the IMU sensor and the Gold Standard show high levels of correlation [61], although there is a lack of consensus with respect to the possible interferences of the device caused by the presence of artifacts that might be considered a source of error, related to the contact with the soft tissue and skin or to the use of the magnetometer to calibrate the gyroscope [62]. Despite this, the IMU sensor is valid and reliable [63] for monitoring spatiotemporal parameters during running [64] and pelvic kinematics, as well as for enabling fast, simple, and affordable assessment [26].

## 5. Conclusions

This study determines the 3D kinematic pattern of the pelvis during running and establishes differences between sexes using an IMU sensor. In particular, an almost 2-degree increase in pelvic obliquity was observed in females at a medium speed level with respect to males, while it remained stable at every other speed level. Pelvic rotation increased by 3.64° at higher speed levels in males, and by 2.92° in females. However, pelvic tilt remained stable at every speed level for both sexes. The inertial sensor has proven to be a reliable resource in the analysis of all spatiotemporal parameters and pelvis kinematics during running.

## Figures and Tables

**Figure 1 ijerph-20-03631-f001:**
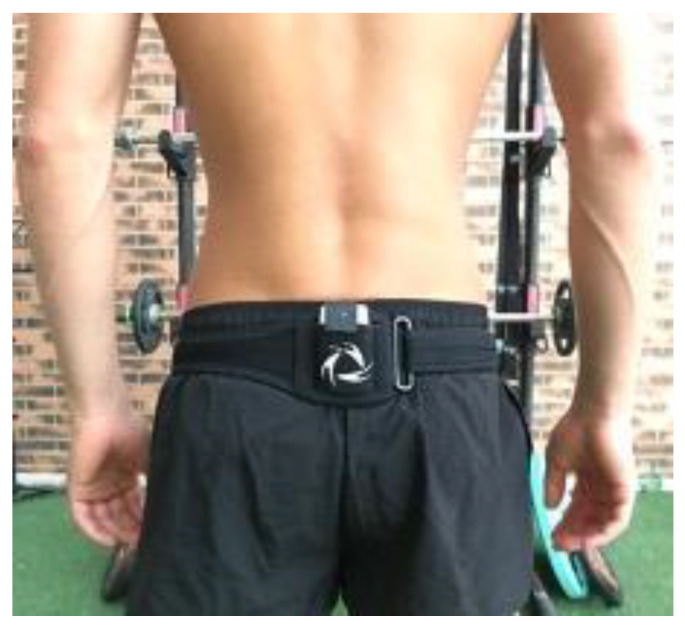
Placement of the IMU in S1.

**Figure 2 ijerph-20-03631-f002:**
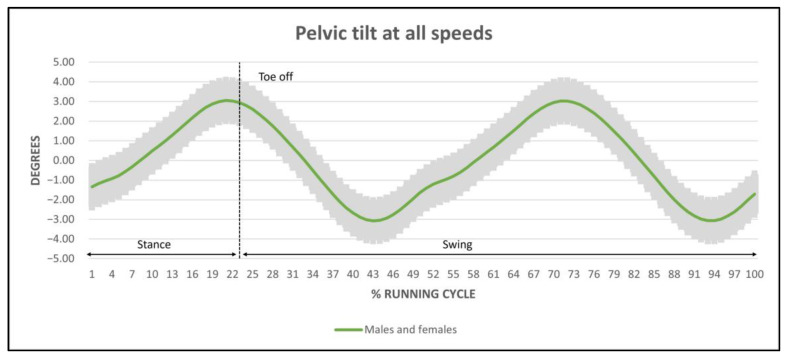
Pelvic tilt trace with mean amplitude in males and females. Negative degrees represent a retroversion of the pelvis. Positive degrees represent anteversion of the pelvis. No statistical differences found in remaining speed levels or between sexes.

**Figure 3 ijerph-20-03631-f003:**
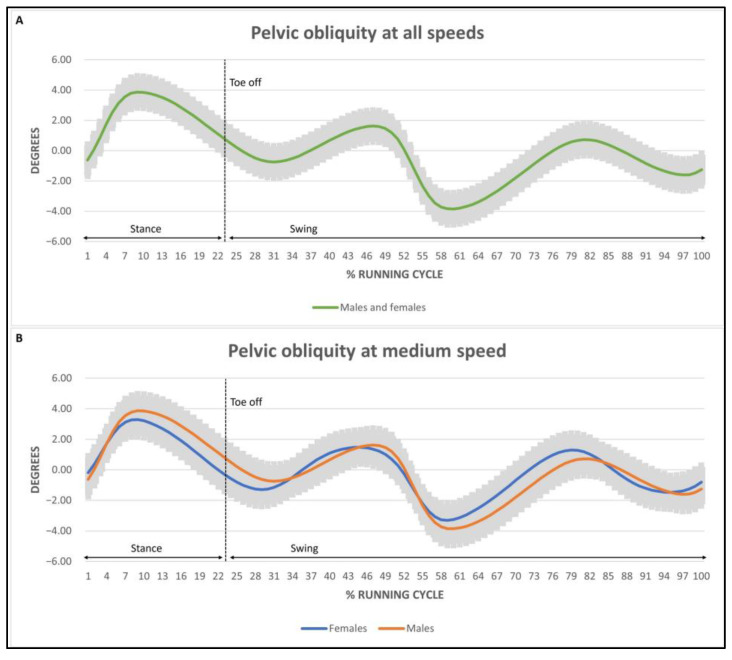
Pelvic obliquity trace at different speed levels with mean amplitude in males and females. (**A**) at all speed levels. (**B**) at medium speed. Negative degrees represent a caudal obliquity. Positive degrees represent an obliquity in the cranial direction.

**Figure 4 ijerph-20-03631-f004:**
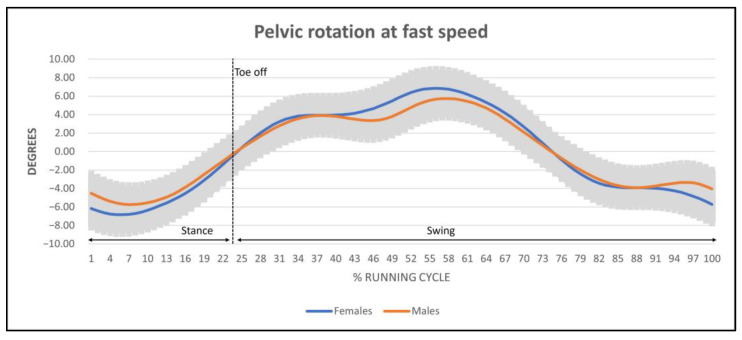
Pelvic rotation traces with mean amplitude regarding the fast speed. Negative degrees represent external rotation of the pelvis, while positive degrees represent internal rotation.

**Table 1 ijerph-20-03631-t001:** Demographic data of participants to establish normative values and sensor reliability according to sex. Values are presented with mean and standard deviation (SD).

	Normative Values		Sensor Reliability	
Outcome	Males	Females	Males	Females
*n*	51	50	14	15
Age (years)	32.49 ± 8.61	30.16 ± 8.94	32.36 ± 9.09	30 ± 8.08
Weight (Kgs)	74.02 ± 6.69	57.25 ± 6.11	74.65 ± 6.69	58 ± 6.09
Height (cm)	176 ± 5.70	165.70 ± 5.79	178 ± 4.94	167 ± 7.02

**Table 2 ijerph-20-03631-t002:** Spatiotemporal outcomes according to sex and speed levels.

	Males (SD)	Females (SD)	Mean Differences (min–max)
	Slow Speed		
Symmetry index (%)	99.04 (0.69)	99.37 (0.34)	−0.33 (−0.80–0.14)
Cadence (p/m)	170.1 (9.7)	171.9 (15.5)	−1.76 (−9.56–6.04)
Stride time (s)	0.70 (0.04)	0.69 (0.06)	0.01 (−0.03 a −0.04)
Stride length (m)	1.81 (0.12)	1.30 (0.14)	0.22 (0.08–0.35)
Pelvic tilt (°)	6.27 (1.70)	6.26 (2.40)	−0.01 (−1.37–1.77)
Pelvic rotation (°)	10.64 (2.47) **	13.21 (2.98) **	−2.57 (−5.09 a −0.05)
Pelvic obliquity (°)	9.27 (2.53)	8.11 (1.34)	1.16 (−1.18–2.51)
	Medium Speed		
Symmetry index (%)	99.43 (0.44)	98.89 (0.73)	0.55 * (0.07–1.03)
Cadence (p/m)	173.0 (10.4)	172.2 (10)	7.81 (−7.02–8.59)
Stride time (s)	0.67 (0.05)	0.69 (0.04)	−0.01 (−0.04–0.02)
Stride length (m)	2.02 (0.23)	1.84 (0.10)	0.18 (0.06–0.31)
Pelvic tilt (°)	5.92 (2.46)	6.57 (2.58)	0.65 (−1.45–1.65)
Pelvic rotation (°)	9.96 (3.76) **	15.74 (3.99) **	−5.78 (−8.26 a −3.30)
Pelvic obliquity (°)	7.84 (2.16)	9.64 (1.77)	−1.80 *(−3.12 a −0.47)
	Fast Speed		
Symmetry index (%)	98.62 (0.82)	98.99 (0.73)	−0.37 (−0.81–0.08)
Cadence (p/m)	171.3 (9.39)	173.5 (10)	−2.117 (−9.47–5.24)
Stride time (s)	0.70 (0.04)	0.69 (0.24)	0.01 (−0.02–0.04)
Stride length (m)	2.47 (0.27)	2.14 (0.14)	0.33 (0.21–0.45)
Pelvic tilt (°)	6.50 (2.00)	7.36 (2.22)	0.86 (−2.38–0.58)
Pelvic rotation (°)	13.60 (3.68) **	16.13 (3.42) **	−2.53 (−4.91 a −0.16)
Pelvic obliquity (°)	8.75 (1.44)	7.81 (1.99)	0.95 (−2.23–0.34)

* Significant difference *p* < 0.025 between groups (Males–Females) ** Significant differences *p* < 0.05 within groups (sex–speed levels).

**Table 3 ijerph-20-03631-t003:** Reliability index for the IMU sensor according to spatiotemporal outcomes, pelvic kinematics, and symmetry index.

	ICC	*P*
Symmetry index (%)	0.808	<0.001
Cadence (p/m)	0.983	<0.001
Stride length (m)	0.997	<0.001
Stride time (s)	0.985	<0.001
Pelvic tilt (°)	0.868	<0.001
Pelvic rotation (°)	0.922	<0.001
Pelvic obliquity (°)	0.963	<0.001

ICC: Intraclass Correlation Coefficient.

## Data Availability

Not applicable.

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
