# Peer review of "Differences between Sexes and Speed Levels in Pelvic 3D Kinematic Patterns during Running Using an Inertial Measurement Unit (IMU)"

_ijerph, 2023, doi:10.3390/ijerph20043631_

Round 1
Reviewer 1 Report
In this work, the authors aimed to understand the 3D kinematic pattern of the pelvis using a 9-Degrees-of-Freedom IMU sensor in running activities. The test was conducted with 101 participants. These authors employed a single sensor under the testing individual's lower back.
Initially, the authors displayed a comprehensive introduction, which builds a good background for context and scientific soundness. They assess the importance of the understanding from kinematic patterns in running activities, considering several references that build up this context. Also, the initial part of the materials and methods section presents a detailed overview of the experimental apparatus. This explanation aids the reproducibility of this work, also contributing with its scientific soundness. Finally, the authors display the methods which they used to perform the analyses. This condition enforces the first part of the results, in which the authors present several results and their initial interpretation. These are the most positive aspects of this text.
There are some small aspects in the writing that can use some improvement. Mostly, the authors should avoid single-phrase paragraphs, and rather merge them with the previous or following paragraph. Overall, the authors provided a considerable amount of information, which has a scientific value by itself. Nonetheless, the understanding of this reviewer is that the article will only be complete when they perform a critical evaluation of the meaning obtained from their results.
In general, the authors could have done a better job in the concluding section.
Reviewer 2 Report
The paper presents very interesting research about using IMU sensor for assessing kinematic parameters of pelvic during running. The whole paper is well written and its worth publication, but In my opinion, should be corrected according to the comments below:
- The title should be one sentence.
- Line 43 - what IT band means.
- The aim used the term: 3 planes of movement, in my opinion it is too general.
- Why the treadmill test was performed at a speed chosen by the subject. Why wasn't a measurement taken for the same person at different speeds. With the assumptions from the paper, we cannot clearly state the differences between speeds because they may be due to individual technique. Using three different speeds for the same person would allow us to determine the effect of speed on kinematic parameters.
- There is a lot of text in the results that should be in the material and methods section. Table 1 is not the results of the experiment but the characteristics of the group. The first paragraph is basically a description that should be in the material. The second paragraph describes the protocols for assigning speeds during the measurement. It should be in the subsection where the research methods are described.
- Table 2 should include information in the footer as to what the value before the parentheses is and the value within the parentheses,
- Figures included in the results are too small, giving the impression of illegibility.
- The text is formatted carelessly. Figure captions extend beyond the margin (see Fig. 2). There are blank lines between paragraphs. There are no dots after sentences (including Line 70 and line 228). Authors should prepare the technical side of the text more carefully.
- Subsection 3.2. consists of only one sentence. In my opinion, it needs more commentary.
- The first two paragraphs are essentially a repetition of the summary of results. What is missing from the discussion is a discussion of the limits of study, and the limits of IMU sensor use. IMU sensors have several limitations related to magnetometer operation or trajectory calculation.
